# Female Leadership during COVID-19: The Effectiveness of Diverse Approaches towards Mitigation Management during a Pandemic

**DOI:** 10.3390/ijerph20217023

**Published:** 2023-11-06

**Authors:** Esra Ozdenerol, Rebecca Michelle Bingham-Byrne, Jacob Seboly

**Affiliations:** 1Spatial Analysis and Geographic Education Laboratory, Department of Earth Sciences, University of Memphis, Memphis, TN 38152, USA; rmbngham@memphis.edu; 2Department of Geosciences, Mississippi State University, Starkville, MS 39762, USA; jds1565@msstate.edu

**Keywords:** female presidents, leadership, mitigation, COVID-19, gender differences

## Abstract

This paper tackles the question of how female leaders at national levels of government managed COVID-19 response and recovery from the first COVID-19 case in their respective countries through to 30 September 2021. The aim of this study was to determine which COVID-19 mitigations were effective in lowering the viral reproduction rate and number of new cases (per million) in each of the fourteen female presidents’ countries—Bangladesh, Barbados, Belgium, Bolivia, Denmark, Estonia, Finland, Germany, Iceland, Lithuania, New Zealand, Norway, Serbia, and Taiwan. We first compared these countries by finding a mean case rate (29,420 per million), mean death rate (294 per million), and mean excess mortality rate (+1640 per million). We then analyzed the following mitigation measures per country: school closing, workplace closing, canceling public events, restrictions on gatherings, closing public transport, stay-at-home requirements, restrictions on internal movement, international travel controls, income support, debt/contract relief, fiscal measures, international support, public information campaigns, testing policy, contact tracing, emergency investment in healthcare, investment in vaccines, facial coverings, vaccination policy, and protection of the elderly. We utilized the random forest approach to examine the predictive significance of these variables, providing more interpretability. Subsequently, we then applied the Wilcoxon rank-sum statistical test to see the differences with and without mitigation in effect for the variables that were found to be significant by the random forest model. We observed that different mitigation strategies varied in their effectiveness. Notably, restrictions on internal movement and the closure of public transportation proved to be highly effective in reducing the spread of COVID-19. Embracing qualities such as community-based, empathetic, and personable leadership can foster greater trust among citizens, ensuring continued adherence to governmental policies like mask mandates and stay-at-home orders, ultimately enhancing long-term crisis management.

## 1. Introduction

This paper tackles the question of how female leaders at national levels of government managed COVID-19 response and recovery. We begin by reflecting on the question that captured public imagination—did female leaders contain the virus more successfully at its early stages? If so, which COVID-19 mitigations did they quickly implement that were effective in containing the virus? Of the 195 independent countries in the world, women led only 14 during COVID-19 pandemic: Bangladesh, Barbados, Belgium, Bolivia, Denmark, Estonia, Finland, Germany, Iceland, Lithuania, New Zealand, Norway, Serbia, and Taiwan. In spite of their diverse cultural backgrounds and environments, did they share common leadership traits distinctive of women in the management and control of the pandemic? The female leadership stories from the early stages of the pandemic from these 14 countries could exemplify this gender perspective and guide research in gendered leadership differences. The goal of this paper was to examine the effectiveness of the measures put in place by these nations’ female leaders, discussing where they succeeded and the countries that did not do well to facilitate a better understanding of female leaders’ success in dealing with COVID at its early stages, which could be helpful for mitigation management for future pandemics. We found that most analyses have focused on proving how female leaders have managed the pandemic better than their male counterparts without trying to identify their mitigation strategies, how these strategies caused the difference in COVID outcomes, and the impact of female leadership and feminine traits on COVID-19 management and control. For this research study, we compared mitigation importance and effectiveness among female-led countries and investigated whether leaders’ gender and traits affected the way their respective countries dealt with COVID-19. In this introductory section, we review the theoretical models of leadership and feminist theory that are particularly pertinent to the discussion of female leaders’ responses to COVID-19 and summarize the findings of various studies that identified gender differences.

### 1.1. A Review of Feminist Leadership Theories

Feminist leadership theories vary in their approaches, from advocating for equal treatment and opportunities (Liberal Theory) to celebrating gender differences (Radical Theory) and analyzing the societal construction of gender roles (Social Constructionist and Post Structuralist Feminist Theory). Each theory offers a different perspective on how women can excel in leadership roles and address gender-related challenges. Liberal theory is grounded in the belief that men and women are fundamentally the same and that gender differences should not be a barrier to leadership or equal opportunities [1]. Women strive for self-improvement and believe they have the same capabilities as men to succeed in various fields, including business or politics. Women aim to challenge societal biases and prejudices that might hinder their progress. Liberal feminists advocate for gender equality by promoting equal opportunities and emphasizing that women can achieve success in leadership roles by acting similarly to men in these positions.

Radical Theory posits that while men and women are equal, they are also different [2]. It celebrates feminine traits and qualities as assets to leadership rather than drawbacks. Women should not try to emulate male leadership styles but should embrace their own unique qualities and traits. Leadership can benefit from the coexistence of distinct elements of male and female leadership, creating alternative styles that appeal to both genders. Radical feminists argue for a reevaluation of leadership qualities and the recognition of the value of feminine leadership attributes in achieving successful outcomes. Social Constructionist and Post Structuralist Feminist Theory shifts the focus from defining what men and women are to examining how society constructs concepts of femininity and masculinity [3]. Masculine traits in leadership are preferred not because they are inherently superior but because society perceives them as desirable. Female leaders can work on challenging and changing negative stereotypes associated with their gender. Social constructionist and post-structuralist feminists emphasize the importance of deconstructing societal norms and expectations related to gender and leadership, with a focus on dismantling biases and stereotypes. 

### 1.2. Research Findings on Gender Differences in COVID-19 Management

There has been controversy over the possibility of female-led countries outperforming male-led countries during the COVID-19 pandemic, beginning with reports in the media stating that female-led countries were outperforming even though some studies have found no statistical differences in the types of mitigation policies used [4]. However, there have been differences associated with the timing of these mitigation policy types [4], the amount of money spent on public health [5], and how the leaders communicate these various mitigation policies [6]. Alam [7] mentions that no single process has been found to be suitable for all countries across the world. There may never be a single process suitable for all countries because people behave and react differently in situations, and the leadership of that area needs to find the best way to manage the crisis for their people. There is no denying the relative early success of leaders such as Germany’s Angela Merkel, New Zealand’s Jacinda Ardern, Denmark’s Mette Frederiksen, Taiwan’s Tsai Ing-wen, and Finland’s Sanna Marin, which attracted many headlines but little academic attention. New Zealand and Taiwan have especially been formidable, being perceived as two of the top three performers in COVID-19 management in 2020, according to a study conducted by the Lowy Institute [8]. Even after clear and frequently cited outliers such as New Zealand and Germany, and the US for male leaders, were removed from the statistics, this study found that the case for the relative success of female leaders was only strengthened.

The timing in which mitigation policies are implemented also have an impact on citizen adherence [9,10,11]. Madeiros and colleagues [12] evaluated COVID-19 management based on governmental leader personality traits (compounded into two “meta-traits”: stability—agreeable with a cautious response—and plasticity—open with a swift response. Some policies may be expected to have a quicker response (limiting contact with others), and others (economic relief) were sometimes slow to start. Citizens’ approval or disapproval with mitigation measures has been found to be related to the timing in which the mitigation protocol was implemented compared to surrounding countries [9]. An analysis of 194 countries published by the Centre for Economic Policy Research and the World Economic Forum suggests the difference between male-led and female-led countries is real and “may be explained by the proactive and coordinated policy responses” adopted by female leaders. The researchers behind this study said they analyzed differing policy responses and subsequent total COVID-19 cases and deaths until 19 May, introducing several variables to help analyze the raw data and draw reliable comparisons between countries. Among the datasets considered by the Center for Economic Policy Research and World Economic Forum were GDP, total population, population density and proportion of elderly residents, as well as annual health spending per head, openness to international travel, and level of gender equality in society in general. Since only 19 of the nearly 200 countries were led by females, the authors also created so-called “nearest neighbor” countries to offset the small sample size, pairing Germany, New Zealand, and Bangladesh with male-led Britain, Ireland, and Pakistan. Countries led by females had “systematically and significantly better” COVID-19 outcomes [13,14,15]. These countries locked down earlier and suffered half as many deaths on average as those led by men. It was stated that female leaders “were risk averse with regard to lives”, locking their countries down significantly earlier than male leaders, which also suggested they were “more willing to take risks in the domain of the economy”. Aldritch and Lotito [4] found no differences in COVID-19 mitigation measures, but they did find differences in the timing and duration in which policies were implemented. According to the *New York Times* [14], women were proactive in restricting movements within their countries, and Aldritch and Lotito [4] pointed out that over half of the female-led countries implemented public information campaigns prior to any reports of COVID-19 within their country. This could be because women were willing to adopt policies quicker when people’s health were at risk [4]. However, this may not be true in all aspects. Female-led countries have also been associated with delays in closing down schools to help ensure higher levels of social and economic status, which was most likely because governments with higher rates of women within them are more aware of the socio-economic costs of closing down schools [4]. When compared according to the “openness to travel” criterion, the authors of [15] found that female-led countries did not experience significantly lower COVID cases but did report lower deaths, concluding that this may suggest “better policies and compliance”. The lower number of COVID cases could also be related to differences in public health spending [5] or in communication to the public, as compliance towards mitigation policies is related to the amount of ‘trust’ a person has in their government [9,16].

Trust in one’s government is essential when dealing with crises, especially in situations where citizens must adhere to special policies for extended periods of time. This is because extended periods of isolation, such as those required during the pandemic, can lead to psychological distress [17], starting with anxiety and leading to depression, obesity, and heart problems [18], as well as other lifelong issues such as post-traumatic stress [19]. A lot of these health issues arise from the uncertainty that emerges, which can be reduced through trust, along with increased adherence to mitigation policies and societal cohesion [20]. In areas where there is a lack of trust in their government, misinformation is spread, and adherence to temporary policies diminishes [21]. In order for people to gain trust in their leaders, transparency of decision-making as well as sharing information with the people is essential; however, the people must also be willing to adapt rapidly to new ideas [22]. How the government uses information and communication technology, such as news platforms and social media, is very important in how well pandemic management works [7,23]. Because of the rapid spread of COVID-19, along with the series of unknowns pertaining to the disease, governments were forced to act fast, without consulting the opinions of the general public [9,24]. Government responses needed to be handled delicately in order to not lower their citizens’ trust. Altimarkis and colleagues [9] mention that polarized communities, such as the United States, have less trust in their government because of opposing political views, which reduces their support towards mitigation measures. Countries in these polarized settings, along with non-polarized countries whose citizens have low levels of trust in their governmental leadership, must take extra precautions in their approach to informing their citizens, especially low-income and vulnerable populations, of how and why to follow emergency protocols. 

It has been shown that the poor were more susceptible to COVID-19 because these individuals had more on-site jobs [22,25], giving them the choice of either losing their jobs or increasing their contact with others. The reduced allowance of on-site jobs led to “catastrophic job loss and economic hardship” for many populations [26], leading to many more issues for these individuals, especially the poor. It increased the already pronounced gap for low-income and minority communities in income, housing displacement, and health care access, which caused them to be more susceptible to COVID-19 and morbidity [26]. It has been noted that women leaders not only prioritized individuals and small businesses in their discussion of the economy, but they also spoke more often about and directly to vulnerable populations, including migrants, refugees, domestic violence victims, and individuals with mental health disorders or substance abuse issues [10]. They also stressed the importance of social welfare and described non-traditional aid such as day care and mental health [10]. These subtle differences in communication can make drastic differences in how low-income and minority communities adhere to mitigation policies. 

While most countries initially had the underlying trust of their citizens due to the “rally around the flag” effect [9,27], governments had to work hard to keep this level of trust throughout the pandemic. This was upheld by appropriately timing the implementation of mitigation measures and effective communication by the governmental leaders. Garikipati and Kambhampati [15] said the evidence of a “significant and systematic difference” showed that even accounting for institutional context and other controls, “being female-led has provided countries with an advantage in the current crisis”. The researchers said they hoped the study would “serve as a starting point to illuminate the discussion on the influence of national leaders in explaining the differences in country Covid-outcomes”. In an effort to adhere to their requests, we examined the differences in mitigation policies in female-led countries during the pandemic by gaining insights into the country-specific importance of the mitigations used and comparing overall mitigation importance and effectiveness among the female-led countries during the pandemic.

## 2. Female-Led Countries

Female political leadership was chosen based on their assumed public gender identity (similar to Dada and colleagues [10]). There were 14 countries that had a female leader throughout the study period of this paper. The names of the female leaders during the pandemic and their respective countries are as follows: Sheikh Hasina (Bangladesh); Mia Mottley (Barbados); Sophie Wilmès (Belgium); Jeanine Áñez (Bolivia); Mette Frederiksen (Denmark); Kaja Kallas (Estonia); Sanna Marin (Finland); Angela Merkel (Germany); Katrín Jakobsdóttir (Iceland); Ingrida Šimonytė (Lithuania); Jacinda Ardern (New Zealand); Erna Solberg (Norway); Ana Brnabić (Serbia); Tsai Ing-wen (Taiwan). All individuals were in office throughout the whole study period unless otherwise specified in the “Tenure notes” section of Table 1 (Table 1). Even if the person was not in office during the entire period, our analysis was conducted considering the entire study period.

We tackled the question of how countries with female leaders at the national levels of government managed COVID-19 response and recovery from the first COVID-19 case in their respective countries from January 2020, until mitigation and rural duration (in days) of mitigation, through to 30 September 2021.

## 3. Materials and Methods

### 3.1. Procedure

We conducted an extensive literature review on feminist leadership theories and the impact of female leadership and feminine traits on COVID-19 management and control. We presented the findings of our literature review in the Introduction, finding that most analyses have emphasized how female leaders managed the pandemic better than their male counterparts without identifying their mitigation strategies, the impact of these strategies on COVID outcomes, and their implication for future political leadership. Given the extensive global database of country profiles, we set out to answer what we consider the following important questions: Why do female leaders seem to be more successful in facing the pandemic? Which female led mitigations were effective in containing the virus quickly? To conduct this analysis, we first identified all the countries that were led by females during the COVID-19 pandemic. We compared these countries by COVID cases and deaths. We then identified their mitigation strategies, pillars, and attributes in which they excelled and subsequently assessed their impact on the country’s overall influence. We compared their mitigation strategies and investigated whether leaders’ gender and traits affected the way the countries dealt with COVID-19. We then applied the random forest approach to assess the predictive significance of the mitigation variables, enhancing result interpretability. Additionally, we conducted statistical analyses to compare when each mitigation measure was in effect and not in effect for the variables that were found to be significant by the random forest model. We also examined the frequency of each mitigation measure found to be significant throughout the models. Later in this paper, future implications for policy guidelines regarding effective mitigation strategies for future health emergencies are mentioned, as are the limitations of this study. 

### 3.2. Data

We identified the countries with female leaders who were successful in managing and controlling COVID-19. We analyzed fourteen female-led countries: Bangladesh, Barbados, Belgium, Bolivia, Denmark, Estonia, Finland, Germany, Iceland, Lithuania, New Zealand, Norway, Serbia, and Taiwan. We downloaded statistics pertaining to country-COVID-19 cases and deaths from WHO’s website [28]. Health data were isolated to include health statistics from a specific time period, starting from when the first case was observed (as early as 15 January 2020) and ending on 30 September 2021.

We also downloaded the mitigation measures of each country from an open-access global database of country profiles called the “Oxford COVID-19 Government Response Tracker” (https://github.com/OxCGRT/covid-policy-tracker accessed on 25 June 2022), which allowed us to explore the statistics regarding the pandemic for every country in the world at a near-real-time rate as data were collected on a weekly basis [28]. Each profile includes an explanation of the presented metrics and details on the sources of the data. The mitigation measures included school closing, workplace closing, canceling public events, restrictions on gatherings, closing public transport, stay-at-home requirements, restrictions on internal movement, international travel controls, income support, debt/contract relief, fiscal measures, international support, public information campaigns, testing policy, contact tracing, emergency investment in healthcare, investment in vaccines, facial coverings, vaccination policy, and protection of the elderly. Data were isolated to include mitigation efforts between a time period spanning from when the first case was observed (as early as 22 January 2020) and to 16 September 2021. A two-week lag was in effect to account for the delays related to viral incubation time and waits for COVID-19 test results. So, for example, the mitigations in effect on January 1 are theorized to be best associated with the health statistics pertaining to January 15.

### 3.3. Data Analyses

In this section, we further describe our data analyses and present the results of our analyses for each female-led country. Our first analysis involved comparing female leaders by COVID cases and deaths using random forest analysis to determine which COVID-19 mitigations were effective in lowering the viral reproduction rate and number of new cases per million in each of the fourteen female-led countries. We further investigated the difference between when each mitigation was in effect and was not in effect for the resulting significant mitigation variables.

#### 3.3.1. Comparison of Female-Led Countries by COVID Cases and Deaths

We first compared the cumulative cases, deaths, and excess mortality rates of the female-led countries at the end of the timeframe studied (Table 2). The countries with female heads of state had a mean case rate of 54,839 per million people, a mean death rate of 733 per million people, and a mean excess mortality rate of +1194 per million people. Barbados (−960), Denmark (−134), New Zealand (−506), Norway (−124), and Taiwan (−194) had negative excess mortality rates, meaning that fewer deaths occurred in that period compared with the baseline period. Belgium (1394), Bolivia (4096), Estonia (1509), Finland (206), Germany (565), Iceland (137), Lithuania (4358), and Serbia (5180) had positive excess mortality rates, meaning that more deaths occurred in that period compared with the baseline period. Bangladesh had no data pertaining to excess mortality, and it was excluded when calculating the mean excess mortality per million. Upon visual examination, countries with higher population densities did not seem to have higher COVID case rates, death rates, or excess mortality rates (Figure 1). 

#### 3.3.2. Random Forest Modeling

In this project, random forest analysis was performed to determine which COVID-19 mitigations were effective in lowering the viral reproduction rate and number of new cases per million in each of the fourteen female-led countries (Bangladesh, Barbados, Belgium, Bolivia, Denmark, Estonia, Finland, Germany, Iceland, Lithuania, New Zealand, Norway, Serbia, and Taiwan). By combining interpretability and flexibility, the random forests produced asymptotically normal predictions, revealing subtle nonlinear relationships and providing a new perspective on our research questions. The utility of this approach on the predictive significance of the mitigation variables enabled us to better interpret the causal inference and delineate the most valuable mitigations by understanding which mitigations influenced the predictive capabilities for differences in the factors used to describe the spread of COVID-19.

Table 3 is a list of the mitigation measures analyzed. Mitigations C1C8 are mitigation measures related to containment and closure policies; E1–E4 all refer to economic policies, and H1-H8 deal with health policies [28].

Each country was represented by a dataset. Each row in the country’s dataset represented a specific date. Each column represented a specific mitigation, and the column contained a 1 if the mitigation was in effect on that date or a 0 if the mitigation was not in effect on that date. Viral reproduction rates and number of new cases per million were also included. The use of the viral reproduction rate rather than raw or normalized case counts reduced the biases associated with past caseloads. If caseloads were high on January 15, they were also likely to be high on January 16, regardless of any mitigations. Reproduction rate controls for this mathematical reality.

#### 3.3.3. Inferential Statistical Analysis of COVID-19 Transmission with Mitigation Measures 

Inferential Statistics Wilcoxon rank-sum test was used to determine if there was a difference in COVID-19 transmission between the times when mitigations were in effect and when mitigations were not in effect. This was used due to the unequal sample sizes and variances between groups. In cases where there were significant differences between groups, the mitigation effect was calculated to determine if the mitigation helped lessen the spread of COVID-19.

## 4. Results

We first examined mitigations by country, starting with days from the first COVID-19 case until the mitigation and the total days of the duration of the mitigation. We further analyzed each individual country in detail by variable importance and assessed the impact of the mitigations that were implemented over the entire study period. Then, we examined and made comparisons of the distributions of variable importance across the seven countries for each of the fourteen mitigation strategies. We also investigated the effects of mitigation on viral reproduction rate (R_0_ of COVID-19) and number of new cases per million. This was carried out by showing whether there was a significant difference in the number of new cases per million or the viral reproduction rate when the mitigation was in effect compared to when it was not in effect. When significant differences were found, the mitigation effects were calculated and visual representations of the mitigation effects were provided.

### 4.1. Mitigations by Country

We were interested in examining the mitigation measures deployed by female-led governments to reduce COVID-19 transmission in their respective countries. We plotted the days from the first COVID-19 case until each mitigation measure for each country (Table 4). We also plotted the total days for the duration of each mitigation (Table 5). This data can be found summarized in Appendix A. We analyzed the COVID-19 mitigation measures per country and the effect of mitigations that female leaders implemented decisively and quickly on to contain the virus. Bangladesh began contact tracing 30 days before they even had their first confirmed case, and they had the shortest length of time between their first confirmed case and their release of vaccines (295 days). Barbados was also quick to act by implementing a few mitigations before they had a confirmed COVID case, including international travel controls (37 days), public information campaigns (41 days), and COVID testing policies (37 days). Bolivia implemented protection for the elderly 56 days before there was a confirmed case within their country. Estonia was the quickest country to provide income support (17 days) after their first confirmed case. However, they were slowest to provide public information campaigns (28 days) and contact tracing (189 days). Finland was the slowest at implementing school closures (63 days), canceling public events (57 days), implementing testing policies (42 days), and implementing mask recommendations/requirements (211 days). Lithuania was the quickest female-led country to act when it came to many of the mitigations presented in the dataset, including school closing (8 days), workplace closing (11 days), canceling public events (7 days), restrictions on gatherings (7 days), closing public transport (11 days), stay-at-home requirements (11 days), restrictions on internal movement (11 days), debt/contract relief (12 days), and fiscal measures (11 days) after their first confirmed case. They also started to provide international support 9 days before their first case, and they were the only female-led country who invested in vaccine development during that time. Norway was the quickest to invest in emergency healthcare (2 days) but the slowest to protect the elderly (295 days). Taiwan was the quickest to implement mask recommendations/ requirements (14 days). However, they were the slowest to close workplaces (126 days), put restrictions on gatherings (126 days), provide income support (102 days), and release vaccinations (437 days). As previously mentioned, countries did not participate in all mitigations throughout the time period. Besides investing in vaccines, here is a list of countries and specific mitigations that they did not use during the timeframe under study: Belgium (international support), Bolivia (international support), Estonia (closing public transport), Finland (closing public transport), Iceland (stay-at-home requirements, restrictions on internal movement, international support, and emergency investment in healthcare), Lithuania (emergency investment in healthcare), New Zealand (international support), Norway (debt/contract relief), Serbia (fiscal measures and international support), Taiwan (international travel controls and international support).

Bangladesh had four mitigation types implemented longer than any other country in the study: closing public transport (502 days), stay-at-home requirements (553 days), restrictions on internal movement (474 days), and international travel controls (618 days). However, they had the shortest terms for income support (31 days) and protection of the elderly (201 days). Belgium provided income support and debt/contract relief for the most days during the study period. Both were in effect for 574 days. Bolivia had more days in which they provided emergency investment in healthcare (7 days), but they had the fewest days wherein testing policies were in effect (542 days). Denmark had the most days with workplace restrictions in effect (569 days). Estonia had the fewest days wherein public information campaigns (568 days) and contact tracing (398 days) were in effect. Germany closed schools for the largest number of days (583 days); also in Germany, the canceling of public events (580 days) and social gathering restrictions (570 days) were in place for the greatest number of days, and Germany also had the most days devoted to protecting the elderly (581 days). New Zealand spent the least number of days with school closures (190 days) and canceled public events (205 days). Norway used more days to provide extra fiscal measures (13 days) and implement vaccine policies (287 days), but the fewest days implementing mask recommendations/requirements (407 days). Taiwan spent the most days providing public information campaigns (638 days), testing policies (620 days), contact tracing (619 days), and mask recommendation/requirements (616 days). However, they had the fewest days implementing workplace closures (139 days), restrictions on gatherings (139 days), and vaccine policies (193 days). Figure 2 provides a visual comparison between the countries in terms of total number of days from first confirmed case until each mitigation measure first started. Figure 3 provides a visual comparison between the countries in terms of total number of days that each mitigation measure was in effect throughout the entire study period.

### 4.2. Comparison across Countries

We were interested in the mitigation measures deployed by female-headed governments to reduce COVID-19 transmission in their respective countries. We analyzed the COVID-19 mitigation measures per country. The boxplot below shows the distribution of variable importance and mitigation effects across the fourteen models (countries) for each of the twenty mitigation strategies.

#### 4.2.1. Comparison of Variable Importance

Facial coverings, vaccination policies, and stay-at-home requirements seemed to be consistently more important across countries for the number of new cases per million model (Figure 4). Debt contract relief, protection of the elderly, contact tracing, restrictions on gatherings, internal movement restrictions, and closing public transport sometimes also seemed to be important, according to the model. However, their mean importance is much lower than the others mentioned. 

Contact tracing, vaccination policy, facial coverings, school closing, and closing public transport seemed to be consistently more important for the viral reproduction rate model (Figure 5). Workplace closing, canceling public events, testing policy, income support, restrictions on gatherings, stay-at-home requirements, restrictions on internal movement, international travel control, debt/contract relief, public information campaigns, and protection of the elderly sometimes seemed to be important, according to the model. However, their mean importance is lower than the others mentioned. 

#### 4.2.2. Comparison of Mitigation Effects

After using the Wilcoxon rank-sum test, we found that there were significant differences in the number of new cases per million (Table 6) or the viral reproduction rate (Table 7) while some mitigations were in effect. Mitigations that were consistently not implemented within the countries or did not have a significant effect on the number of new cases per million or viral reproduction rate included economic or health investment factors, including the following: fiscal measures, international support, emergency investment in healthcare, and investment in vaccines. Because these mitigations did not have much effect on the spread of the disease for most countries, they were omitted from calculations and visualizations regarding the mitigation effects.

The x-axis numbers refer to the difference between the health statistic (either number of new cases per million or viral reproduction rate) when the mitigation was in effect versus the reproduction rate when the mitigation was not in effect. Thus, positive values mean that the mitigation in question was associated with increased COVID-19 transmission, while negative values mean that the mitigation was associated with decreased COVID-19 transmission (i.e., it was effective). Limitations on internal movement and public transportation closures seemed to be more effective across countries in reducing the number of new cases per million (Figure 6) and the viral reproduction rate (Figure 7).

## 5. Discussion

We hope that this study will “serve to help illuminate the discussion on the influence of female national leaders in explaining the differences in country Covid-outcomes”. In this section, we discuss the efficiency of the aforementioned female-led countries in containing the virus and mitigating deaths during the pandemic. We also address the countries individually in relation to their mitigation measures before ending the section by discussing the importance and understanding of the mitigation measures used during the pandemic. Overall, this section discusses the efficiency of the leaders of each country, with a focus on feminist leadership theories. 

### 5.1. Which Countries Did Better?

Even though some countries had high population densities, they may not have had high numbers of COVID-19 cases and deaths or higher excess mortality rates. The countries that “did better” in the context of COVID are those who had fewer cases. The top five countries in this regard include Taiwan (680 cases per million), New Zealand (836 cases per million), Bangladesh (9187 cases per million), Finland (25,670 cases per million), and Barbados (29,804 cases per million). Bangladesh had the highest population density (1278 people per square kilometer); Taiwan was second highest (673.7 people per square kilometer), and Barbados was third highest (669.1 people per square kilometer). With this in mind, it is important to note that the leaders of these three countries did exceptionally well compared to the other female-led countries in controlling the number of emerging cases. The lowest 5 countries in this regard include Serbia (137,085 cases per million), Lithuania (119,338 cases per million), Estonia (117,601 cases per million), Belgium (107,218 cases per million), and Denmark (61,288 cases per million). However, just because a country did not do well controlling the spread of the disease does not mean that they were ineffective in pandemic management.

Part of controlling a pandemic is confirming cases and medicating those who have contracted the disease to help mitigate deaths. Countries that “did better” in mitigating deaths include those whose deaths per million were lower, as well as those whose excess mortality rate remained low. The top five countries with the lowest death rates include New Zealand (6 deaths per million), Taiwan (35 deaths per million), Iceland (89 deaths per million), Norway (159 deaths per million), and Bangladesh (162 deaths per million). Furthermore, the top five countries in maintaining excess mortality rates include Barbados (−960 per million), New Zealand (−506 per million), Taiwan (−194 per million), Denmark (−134 per million), and Norway (−124 per million). Belgium had the highest death rate (2205 per million), followed by Lithuania (1793 per million), Bolivia (1551 per million), Serbia (1198 per million), and Germany (1123 per million). Serbia also had high excess mortality of 5180 per million. This was the highest among the female-led countries, followed by Lithuania (4358 per million), Bolivia (4096 per million), Estonia (1509 per million), and Belgium (1394 per million). There was no information about excess mortality rates in Bangladesh. Denmark had negative excess mortality rates, even though they had higher than average rates of COVID-19 cases. Given the above-mentioned information, it can be noted that the leaders in Bangladesh, Barbados, Denmark, New Zealand, and Taiwan did well in mitigating deaths within their countries.

Looking at overall management of COVID-19 based on the health statistics described above, five countries stand out: Bangladesh, Barbados, Denmark, New Zealand, and Taiwan. These countries effectively kept low rates in at least two of the three health statistics. Gross domestic product (GDP) rank and total funds in the international monetary fund (IMF) for the years 2019 and 2020 are readily available on the Statistics Times website (https://statistictimes.com/economy/countries-by-gdp.php accessed on 25 June 2022) [29]. In 2020, Bangladesh was ranked 41st in GDP, with a total of USD 329,120 billion in their IMF, and its economy was growing until the pandemic emerged [7]. A recent paper mentioned that they are the 8th most populated country in the world [7]; most of the nation’s populace, according to the Central Intelligence Agency [30], are of Bengali ethnicity (98.9%), and the predominant religion is Muslim (88.4%). Only about 40.5% of the total population lives in urban settings, and even though there are more male youths, the total population has slightly more females than males (male to female sex ratio = 0.96) [30]. Barbados was ranked 155th for GDP, with USD 4365 billion in their IMF. The majority of Barbados’ inhabitants are of African descent (92.4%), and about 66.4% of individuals within the country are Protestant [30]. Even though this country is the most densely populated within the Eastern Caribbean, only 31.4% of individuals live in urban settings [30]. Similar to Bangladesh, there are slightly more females than males when looking at the total population (male to female sex ratio = 0.93) of the country [30]. Denmark was ranked 36th in GDP, with USD 652.243 billion in their IMF. According to the CIA [30], most individuals within the country are Danish (85.6%), and the predominant religion is Evangelical Lutheranism (74.7%). Unlike Bangladesh and Barbados, most individuals in this country live in an urban setting (88.5%), and the sex ratio is very close to equal (male to female sex ratio = 0.99) within the total population [30]. New Zealand was ranked 50th in GDP, with USD 209.329 billion in their IMF. Most individuals living in New Zealand are of European descent (64.1%), with the majority of individuals not practicing a religion (48.6%), according to a 2018 census compiled for the CIA [30]. The same census found that 37.3% of the population practiced Christianity. Like Denmark, most of the population can be found living in urban settings (87%), and there is an equal representation of males and females (male to female sex ratio = 1.00) within the total population [30]. Finally, Taiwan was ranked 21st in GDP, with USD 668,510 billion in their IMF. Most individuals living in Taiwan are Han Chinese (>95%), and the majority of people living within the country practice either Buddhism (35.3%) or Taoism (33.2%) as their religion [30]. Most of the population lives in urban settings (80.1%), and there are slightly more females than males (male to female sex ratio = 0.97) within the total population [30].

Countries that did not do well include Belgium, Lithuania, and Serbia. In 2020, Belgium was ranked 25th in GDP, with a total of USD 513.087 billion in their IMF. The majority of Belgium is of European descent, specifically Belgian (75.2%), and about 57.1% of individuals practice Roman Catholicism as their religion [30]. An overwhelming majority of the population (98.2%) live in urban settings [30]. There are slightly more females than males in the total population of the country (male to female sex ratio = 0.97) [30]. Lithuania was ranked 82nd in GDP, with USD 55.688 billion in their IMF. The majority of Lithuania are Lithuanian (85.3%), and about 74.2% of individuals within the country practice Roman Catholicism as their religion [30]. Most individuals (68.7%) live in urban settings [30]. Regarding the total population of the country, there are more females than males (male to female sex ratio = 0.86) [30]. Serbia was 83rd in GDP with USD 52,960 billion in their IMF. The majority of Serbia’s inhabitants are Serbs (83.3%), and about 84.6% of individuals practice the Orthodox religion [30]. Slightly more than half (57.1%) of the population live in urban settings [30]. There are slightly more females than males in terms of the total population of the country (male to female sex ratio = 0.95) [30]. In order to better understand the role these countries had in pandemic management, we will discuss the countries’ roles and the mitigation measures used during this time.

### 5.2. Mitigation Measures

As shown in our analysis, the mitigation measures that were consistently important across countries included mask mandates, vaccination policies, and stay-at-home requirements. These measures were also found to be effective in reducing the number of new cases emerging. Mitigation measures requesting or requiring people to use face masks are a very useful preventative tool when dealing with airborne viruses because it helps to reduce contact rates [11,31,32,33]. Taiwan was the quickest of the countries in the analyses to implement facial covering measures after their first case emerged (14 days), followed by Barbados (39 days). These make up two of the five countries that were found to do well in managing the pandemic in our study. The rest of the countries waited much longer before implementing this measure. Taiwan also actively implemented facial covering measures for the longest duration throughout the study period (616 days). In fact, Taiwan had stockpiled both surgical and N95 masks before the first case was reported in preparation for the pandemic [23,34] and produced more even after the first case, allowing for a surplus to be supplied to other countries [34,35,36,37]. Their preemptive measures and efficiency in utilizing this type of mitigation is not surprising given that people within this country, along with Japan and China, have been utilizing face masks for various purposes, from keeping their faces warm during cold months to protecting against harmful airborne particles, for over 60 years [37]. Taiwan’s strict regulations and the general population’s adherence to mask use is especially interesting when looking at the fact that they were fairly slow at implementing vaccine policy measures (437 days) and stay-at-home policies (130 days) compared to other countries that did well. 

**Vaccine policies** were related to who was able to access the vaccine and when—usually starting with the elderly and vulnerable populations and then including adults before a vaccine became available rest of the general population. “COVID-19 vaccines help our bodies develop immunity to the virus that causes COVID-19 without us having to get the illness” [38]. The more people who become immune to the virus, the closer the population is to achieving “herd immunity”, lessening the likelihood of new cases emerging [39]. This mitigation measure is a little more difficult to understand with respect to the swiftness and duration of implementation because a vaccine for the disease was not fully developed until much later in the study period, when most countries had already tweaked their COVID-19 management plans to ensure they worked best for them. The final “important and effective” mitigation measure we found was stay-at-home policies. This mitigation measure was widespread and has been researched in detail because of the detrimental effects it can have on a person’s mental and physical well-being [18]. However, when the general public fails to adhere to social distancing suggestions provided by the government and health officials, stay-at-home mandates may become an effective preventative measure, as it forces limited social contact, thereby reducing the chance of a person being exposed to the virus [40,41,42]. McNeil [43], as well as Aldrich and Lotito [4], noted how important this mitigation measure was in controlling the spread of COVID-19. Most countries that did well implemented stay-at-home mitigation measures within 40 days of the first observed case, except Taiwan, which waited 130 days before implementation. This could be because less cases were emerging due to high mask mandate compliance. 

Other mitigation measures that reduced social contact, such as school and work closures, public event cancelations, social gathering restrictions, international travel controls, and the protection of the elderly, were effective in controlling the number of new cases emerging but were not as important within the predictive models. The role of school closures in COVID-19 prevention is complicated. Head and colleagues [44] discussed how elementary students still have increased risk to exposure when schools are closed because they must either accompany their parents or attend daycare and how school closures can be detrimental to the mental and social development of these individuals, whereas high school students can be left at home without supervision and are more capable of understanding at-home learning. Aldrich and Lotito [4] mention that female leaders are more aware of this issue, which is why areas with women in leadership often delayed closing schools. Aldrich and Lotito [4] also mention that school closures affect adult women more than men as they are likely the ones who must stay at home to care for the children when they are not in school. Ebrahim and colleagues [16] mention that school closures must coincide with work closures, along with providing distance learning and meal options, especially for the poor, for better efficacy. Work closures can also have bittersweet impacts on individuals. On one hand, they are now less likely to be exposed to COVID, but these individuals are also now less likely to have an income. Lower levels of work closures could include rotating schedules, adding in new shifts, video conferencing, and work-from-home options [16]. Working from home may reduce COVID-19 transmission and the economic hardships associated with it; however, when people work from home, they must have some sort of hands-on learning capability [45]. Lithuania was the quickest to close schools and workplaces, with workplaces being closed within 3 days of the country’s school closure measures coming into effect. Taiwan was the last country to implement workplace closures after the first case emerged. Ebrahim and colleagues [16] describe the importance of social gathering restrictions even when viral reproduction is low because of how much these events increase social contact. Again, Lithuania was quick to implement this type of mitigation measure, with Taiwan being the last country to instill this type of regulation. This was probably because they had such extensive mask wearing measures and good adherence to these measures.

**International travel controls** are extremely important to reduce the spread of airborne diseases such as COVID [16,46,47], and Alam [4] mentions that this is the first step to reducing their spread. Many countries within our analysis instilled international travel regulations before the first case emerged, such as Bangladesh, Barbados, Iceland, New Zealand, and Serbia. Belgium and Germany were late in implementing this type of mitigation. Our analysis showed that what may have been even more important in reducing the spread of COVID-19 was restrictions on internal movement and closing public transportation, which can be related back to international travel controls as it limits how far and where people within a country can travel, thus limiting the amount of people individuals have contact with and making them less likely to contract the disease (if they do not have it) or pass on the disease (if they do). Lithuania was the fastest in both closing public transport and putting restrictions on internal movement. Denmark closely followed Lithuania by also quickly putting restrictions on internal movement in place. Estonia and Finland never implemented public transportation closures, and Iceland never implemented restrictions on internal movement. Germany was one of the slowest countries to close public transportation. Taiwan was not only slow at closing public transportation but also slow to impart internal movement restrictions within their country. It is important to **protect the elderly** and vulnerable populations because they are at the highest risk of hospitalization and death [48]. Bolivia and Bangladesh were proactive in implementing safety regulations for the elderly, while Taiwan and Norway waited the longest after their first cases emerged before implementing a measure related to this.

Economic relief (income support and debt/contract relief) and preventative measures not related to direct social isolation (public information campaigns, testing policy, contact tracing) were also effective management tools but not as consistently important according to the predictive models we used. Governmental measures received more support and adherence in areas where economic measures were more generous [9]. Income support and debt/contract relief was key in trying to maintain our way of life. According to Amis and Janz [22], COVID-19 contributed to massive unemployment and public debt and also caused issues related to food security and increased income and health inequities. It also increased issues with pre-existing housing displacement crises in some areas, making these individuals more at risk of exposure [26]. However, there has been controversy over the best way to implement economic relief [49]. Anderson and colleagues [50] pointed out that it is highly unlikely that government administrations will be able to both minimize public health and economic impacts due to the pandemic. They, along with Hollingsworth and colleagues [11], mention that epidemiologists that help policymakers must decide where their priorities stand amidst crises such as a pandemic. In these circumstances, policymakers must trust their scientific advisors on how to properly manage the pandemic, and this trust is essential in policy design [20]. It is easier for a policymaker to trust scientific advisors when they themselves have a scientific background, like the German Chancellor, Angela Merkel, who was proactive in responding to the pandemic [51], grasped the seriousness of the disease, and implemented policies quickly [24]. It is important for the public to understand how mitigation policies were developed, as the behavior of each individual within society has an effect on the spread of the disease [20,50]. 

**Public information campaigns** played a huge role in COVID-19 management, as they allowed governmental administrations to relay information to the public about the disease and how it was being managed. These types of campaigns also help governmental administrations build trust with their citizens, which should lessen the amount of misinformation spread throughout society, and influence what makes sense to the general population [52]. This allows the government and scientific advisors to be transparent in their decision making. According to Jarman and colleagues [53], “transparent and competent scientific advice can also improve intergovernmental coordination”. While citizens may be very supportive of mitigation measures in the beginning, governmental administrations may lose support, especially if their policy responses were mismanaged [54] or the citizens begin to feel the effects of economic hardship and lockdown fatigue [9]. These effects can lead to higher stress levels among individuals, which then triggers stress responses within a community (e.g., causing people to hoard irrelevant items and hindering authorities in allocating essential resources to the public) [18]. Public information campaigns can help keep the public supportive of mitigation measures by explaining the reasoning behind why they are in place. Public information campaigns can also be beneficial in addressing misinformation and monitoring the spread of the disease during a pandemic, especially when they are carried out utilizing different communication platforms, including social media, news, leaflets, etc. [7,23]. Social media platforms can be used to effectively alleviate stress from social isolation and improve the overall mental health of the public [18]. Bangladesh had issues with properly utilizing social media platforms to help alleviate the spread of misinformation and stress within their population [7,55]. How the disease is viewed by leaders and the timing of mitigation measures are usually detailed in public information campaigns [9]. It has been noted that the leaders of the US and Brazil during the pandemic compared COVID-19 to the flu, which was reflected in both how they talked about COVID during public information campaigns and how quickly they responded with mitigation measures. Italy and New Zealand treated the disease with more caution and quickly responded with mitigation measures [9]. Altiparmakis and colleagues [9] also mention that the people in Germany, Australia, and Sweden also had issues with their government’s response to the pandemic. When political leaders brush off the seriousness of a disease, they can lose their citizen’s trust. On the other hand, Taiwan was very active in relaying public announcements dedicated to the importance of wearing masks [34]. Furthermore, female leaders were especially alert to the effects of the pandemic on individuals and focused their speeches on individual levels, relating back to important issues for individuals, such as social inequities, economic hardship, and vulnerable populations [10]. Examples that Dada and Colleagues [10] mention include the German Chancellor’s speech in March 2020, which centered around the economic impact the pandemic was having on individuals, and how the leaders of New Zealand and Belgium mentioned that immigrants and refugees were at high risk for contracting the disease.

**Testing policies** are important for identifying people who have contracted the disease, and they work well in conjunction with contract tracing to monitor the spread of a disease during a pandemic. Contract tracing was a key tool in reducing COVID-19 transmission during the pandemic, seeing as both pre-symptomatic (infection detected before symptoms begin) and (infection detected but symptoms never develop) individuals are able to transmit the virus to others [56]. A recent study affirmed that when these two mitigation policies were used in conjunction, there was a substantial reduction in the number of new cases per day [57]. This, of course, only occurs when it is used properly. Many countries successfully deployed these strategies to reduce COVID-19 transmission, including South Korea, Singapore, and China [58,59]. Taiwan was also very successful in using contact tracing to identify and isolate infected individuals sooner [60]. However, the United States and United Kingdom were less successful utilizing these mitigation measures [59]. Some reasons attributed to their reduced impact on COVID-19 transmission include the failure of infected individuals to supply all contacts within the given timeframe, the failure of officials to reach the infected person’s contacts, and infected individuals’ unwillingness to comply with quarantine orders [60]. By identifying and incorporating successful mitigation techniques elsewhere, countries could have improved their own efforts in reducing COVID-19 transmission via the use of proper contact tracing and COVID-19 testing practices. Australia is an example of a nation that managed this. The country revamped their protocols regarding these mitigation types after a big outbreak forced them to reinstate lockdown measures, but they did not have the chance to test their new protocols because the lockdown measures reduced community transmission [60].

## 6. Conclusions

For this study, we conducted a comparative analysis of pandemic mitigation strategies in countries led by women during the pandemic. We explored the presence and influence of female leadership in diverse national contexts, considering the traits of female leaders in their mitigation strategies and focusing on feminist leadership theories.

Gender’s role in leadership is multifaceted, often intersecting with factors like culture, socioeconomics, and historical contexts. These factors, alongside gender, shape how female presidents operate and influence outcomes. While our analysis reveals performance variations among countries with female presidents, it does not dismiss the significance of gender-related considerations. Our findings highlight that gender is not the sole influencer, but there are common parameters associated with female leadership, such as decisive and timely action, clear communication, and risk-averse decision-making.

Female leaders like Bangladesh’s, Barbados’s, Denmark’s, New Zealand’s, and Taiwan’s presidents demonstrated exceptional crisis management qualities. They embraced a detail-oriented, empathetic, and adaptable leadership style, challenging traditional gender stereotypes. These leaders excelled in conveying information, showing empathy, providing clear and transparent communication, prioritizing collaboration, and adapting swiftly. They prioritized science and public health over politics and effectively conveyed the severity of the situation without causing undue panic. For example, Angela Merkel’s (Germany’s Chanceller during the study period) background in science and her pragmatic, data-driven leadership aligned with radical feminist leadership theories that emphasize women’s meticulous and detail-oriented traits. New Zealand’s Jacinda Ardern’s success in controlling the virus is often attributed to her empathetic and compassionate leadership style. Her approach aligned with radical feminist leadership theories emphasizing women’s capacity for empathy and social welfare prioritization, celebrating feminine traits and qualities as assets to leadership rather than drawbacks. Her effective use of clear communication and community well-being initiatives reflects the idea that female leaders may be more inclined to focus on the welfare of their citizens during crises. Taiwan’s Tsai Ing-wen’s efficiency in managing the pandemic can be linked to her adaptability and resilience. Her leadership is indicative of a proactive approach, aligned with social constructionist and post-structuralist feminist theories suggesting that female leaders often exhibit adaptability and an ability to pivot in response to changing circumstances [3].

Learning from their communication skills can enhance crisis leadership, improve public compliance, and build trust. While gender’s influence on leadership is multifaceted, this study focuses on the immediate responses of female leaders during the pandemic’s peak, offering valuable insights into crisis management and leadership.

Notably, our study emphasizes that both developed and developing countries can perform well during a pandemic crisis under female leadership. It underscores the principles of liberal feminism theory, promoting gender equality and challenging stereotypes about women’s leadership capabilities. The Bangladeshi president’s accomplishments align with liberal feminist theory. By holding the highest office in a traditionally patriarchal society, she challenged stereotypes that suggest women are not fit for leadership or decision-making roles. Her success demonstrates that women can excel in leadership positions, including managing complex crises like a pandemic.

This is similar to previous research [7,61] related to overall pandemic management. This shows that proper leadership within the country is essential to the success of pandemic management. Alam [7] discusses the multi-dimensionality of leading a population and how each aspect (political, administrative, and civic) plays an important role in effective crisis management, especially administrative leadership for developing countries [62]. 

Highlighting the successful approaches of these female leaders could lead to a potential shift in the way we view basic values that are usually ascribed to sexual/gender differences. Furthermore, the personalities and values of a leader will affect what that individual does while in office [12,63,64,65], as well as impact the governmental policies for the community [12,66,67]. By successfully managing the pandemic, the female presidents challenged deeply ingrained cultural norms and beliefs about women’s capabilities. Their leadership could contribute to a gradual shift in societal attitudes, leading to more gender-inclusive and equitable practices.

### 6.1. Future Implications

This study on successful pandemic management under female leadership holds significant implications for governance, gender equity, and global public health. It underscores the importance of gender-inclusive leadership, potentially inspiring greater gender diversity in decision-making roles.

These implications may encourage women to pursue leadership careers, resulting in more equitable representation and shifts in policy priorities. Governments may adopt more consultative decision-making approaches, promoting collaboration and inclusivity.

On an international level, effective pandemic management by female leaders can lead to increased cooperation among countries in responding to crises and improve long-term pandemic preparedness. It may also challenge traditional gender stereotypes and contribute to a more equitable distribution of domestic responsibilities.

Moreover, this study’s findings can advance global health equity by focusing on vulnerable populations. Future research on female leadership effectiveness and enhanced gender equality in educational and leadership programs can cultivate leaders who prioritize inclusivity and collaboration.

In future studies, the effectiveness of mitigations regarding reducing hospitalizations could be included in the model to assess whether mitigation strategies that may not reduce the overall spread of COVID-19 can at least reduce the number of severe cases.

### 6.2. Limitations of the Study

This study focused on assessing the efficacy of the mitigation strategies employed by female leaders during the pandemic to reduce viral transmission. However, this study did not account for the impact of these strategies on hospitalizations and deaths. While it is reasonable to assume that reduced cases may lead to fewer hospitalizations and deaths, we did not draw direct conclusions regarding these outcomes.

Additionally, for the present study, we did not consider the broader societal costs associated with COVID-19 mitigation measures, including economic downturns, the loss of educational opportunities for children due to school closures, mental health issues arising from isolation, and other social and economic consequences. A comprehensive discussion of appropriate COVID-19 mitigation strategies should encompass these broader societal impacts alongside analyzing transmission reduction.

Moreover, as with all studies involving government health data, in the production of this manuscript, we faced limitations stemming from inconsistencies in data collection and reporting. Variations in testing rates, diagnostic accuracy, and reporting standards across countries can lead to an incomplete and sometimes inaccurate depiction of the pandemic’s true extent.

Reporting delays, the absence of comprehensive demographic information, data sourced from multiple agencies with varying standards, potential political influences on data reporting, changes in testing methods, the underreporting of asymptomatic cases, decreasing data accuracy over time, and privacy concerns all contribute to the complexity and limitations of interpreting COVID-19 data. Therefore, a cautious approach is crucial when interpreting and analyzing such data, considering these inherent constraints and the broader context.

## Figures and Tables

**Figure 1 ijerph-20-07023-f001:**
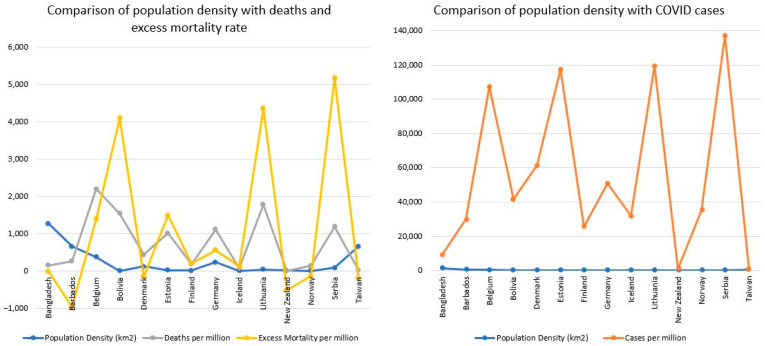
Comparison of population density with health statistics in female-led countries during the study period (1 January 2020–30 September 2021). The graph on the right shows the cases per million and population density for each country. The graph on the left shows the death rates, excess mortality rates, and population density for each country.

**Figure 2 ijerph-20-07023-f002:**
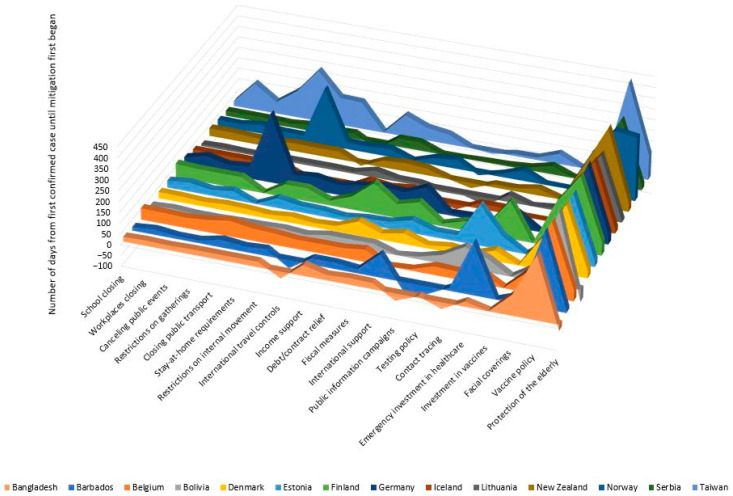
Summary of COVID-19 mitigation efforts in female-led countries during the pandemic in terms of total number of days from first confirmed case until the initial implementation of each mitigation measure.

**Figure 3 ijerph-20-07023-f003:**
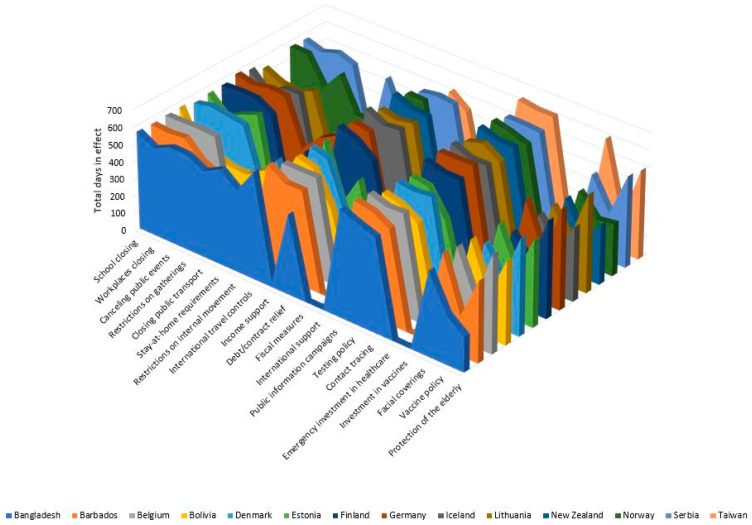
Summary of COVID-19 mitigation efforts in female-led countries during the pandemic in terms of the total number of days that each mitigation measure was in effect for throughout the entire study period (1 January 2020–30 September 2021).

**Figure 4 ijerph-20-07023-f004:**
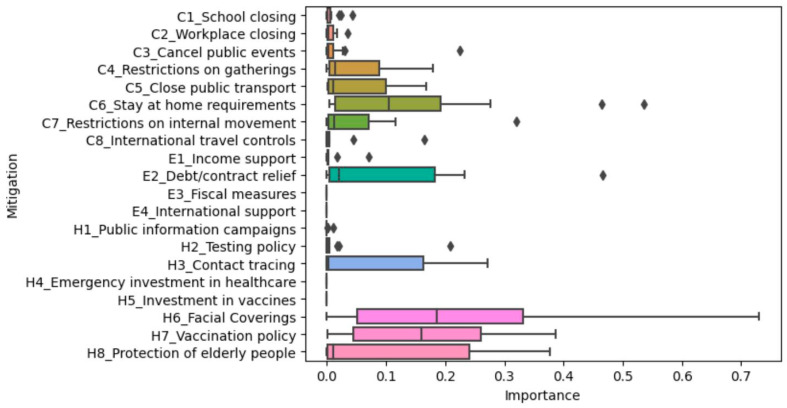
The distribution of variable importance across the fourteen countries with respect to number of new cases per million. The 
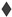
 represents outliers.

**Figure 5 ijerph-20-07023-f005:**
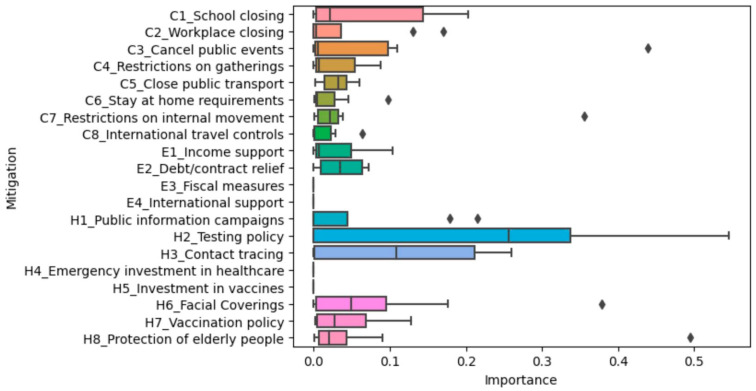
The distribution of variable importance across the fourteen countries with respect to viral reproduction rate. The 
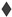
 represents outliers.

**Figure 6 ijerph-20-07023-f006:**
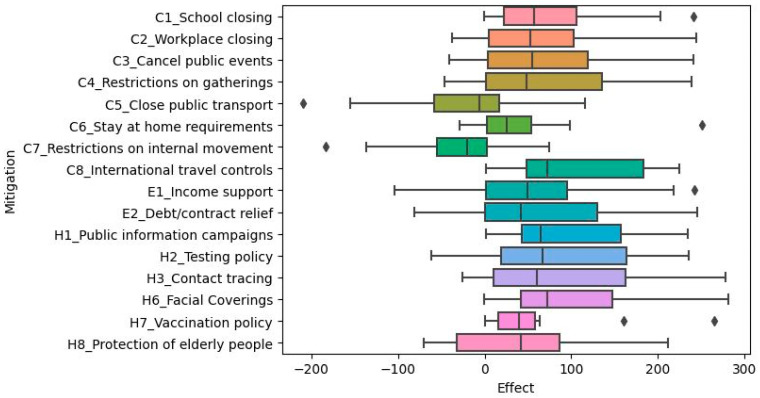
The distribution of mitigation effects across the fourteen countries with respect to number of new cases per million. The 
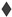
 represents outliers.

**Figure 7 ijerph-20-07023-f007:**
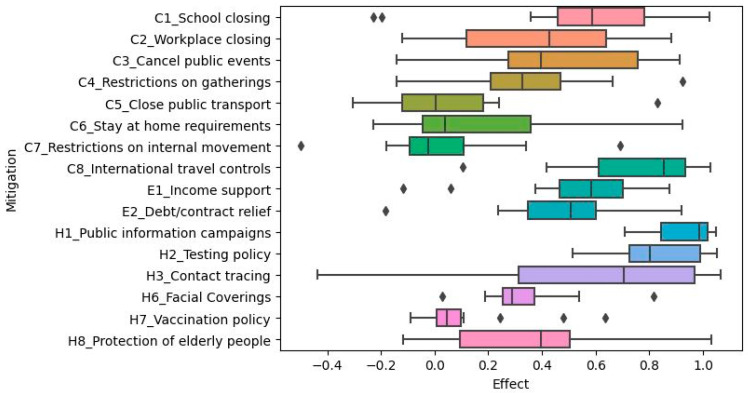
The distribution of mitigation effects across the fourteen countries with respect to viral reproduction rate. The 
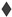
 represents outliers.

**Table 1 ijerph-20-07023-t001:** The females who served as leaders of specific countries during the COVID-19 pandemic (January 2020–September 2021).

Name	Country	Tenure Notes
Sheikh Hasina	Bangladesh	
Mia Mottley	Barbados	
Sophie Wilmès	Belgium	Exited office October 2020
Jeanine Áñez	Bolivia	Exited office in November 2020
Mette Frederiksen	Denmark	
Kaja Kallas	Estonia	Entered office January 2021
Sanna Marin	Finland	
Angela Merkel	Germany	
Katrín Jakobsdóttir	Iceland	
Ingrida Šimonytė	Lithuania	Entered office November 2020
Jacinda Ardern	New Zealand	Won reelection in October 2020
Erna Solberg	Norway	
Ana Brnabić	Serbia	Reappointed in October 2020
Tsai Ing-wen	Taiwan	

**Table 2 ijerph-20-07023-t002:** Outcomes by country.

Country	First Case	Cases per Million	Deaths per Million	Excess Mortality per Million	Population Density (km^2^)
**Bangladesh**	23 February 2020	9187	162	N/A	1278
**Barbados**	3 March 2020	29,804	263	−960	669.1
**Belgium**	21 January 2020	107,218	2205	1394	384.2
**Bolivia**	26 February 2020	41,429	1551	4096	10.92
**Denmark**	13 February 2020	61,288	454	−134	137.0
**Estonia**	13 February 2020	117,601	1021	1509	31.26
**Finland**	15 January 2020	25,670	202	206	18.26
**Germany**	13 January 2020	50,684	1123	565	240.7
**Iceland**	14 February 2020	31,866	89	137	3.425
**Lithuania**	5 March 2020	119,338	1793	4358	42.92
**New Zealand**	14 February 2020	836	6	−506	18.46
**Norway**	12 February 2020	35,060	159	−124	14.96
**Serbia**	21 February 2020	137,085	1198	5180	99.45
**Taiwan**	10 January 2020	680	35	−194	673.7

N/A stands for no associated data.

**Table 3 ijerph-20-07023-t003:** Mitigation measures.

Mitigation Measures
**School closing (C1)** relates to the closure of various educational facilities (schools and universities), ranging from no measures to the required closure of schools of all levels (which led to virtual learning).
2. **Workplace closing (C2)** relates to the various workplace closures, ranging from no measures to the required closure of all workplaces (which led to working from home), except essential workplaces (e.g., grocery stores, clinics, hospitals).
3. **Canceling public events (C3)** relates to the varying levels of public events cancelation and banning, ranging from no measures to the required cancelation of all events outside of the “critical” sectors and “essential” services.
4. **Restrictions on gatherings (C4)** relates to the varying levels of restrictions in the number of people that are allowed at gatherings or events for social, communal, spiritual, religious, recreational, leisure, and sporting purposes. The levels range from no restrictions to restrictions on gatherings of 10 people or less.
5. **Closing public transport (C5)** relates to the varying levels of public transportation closures, ranging from no measures to the required closing of public transport (or prohibiting most citizens from using it).
6. **Stay-at-home requirements (C6)** relates to the varying levels of “shelter-in-place” and other types of home confinement measures set by the governments, ranging from no measures to requiring people to not leave the house, with minimal exceptions (e.g., once a week or only one person at a time.)
7. **Restrictions on internal movement (C7)** relates to the varying levels of restrictions on movement between cities/regions, ranging from no measures to internal movement restrictions being in place.
8. **International travel controls (C8)** relates to the varying levels of restrictions on foreign travelers being allowed to enter the country, ranging from no restrictions to bans on all regions or total border closures.
9. **Income support (E1)** relates to the varying levels of financial aid provided by the government as direct cash payments for those who lost their jobs or were unable to work because of COVID-19. The levels range from no income support to the government replacing 50% or more of lost salary (or if a flat sum is provided, the amount was usually greater than 50% of the median salary).
10. **Debt/contract relief (E2)** relates to the varying levels of government-mandated freezing or stopping financial obligations for households (e.g., loan repayments, utilities, evictions), ranging from no debt/contract relief to broad debt/contract relief.
11. **Fiscal measures (E3)** relates to the amount of new spending or tax cuts that were not already included in any of the other mitigation measures provided, ranging from no spending to amount (in USD) spent that day.
12. **International support (E4)** relates to the amount of money (USD) given to support to other countries during the pandemic.
13. **Public information campaigns (H1)** relates to the varying levels of COVID-19-related information dispersed to the general public, ranging from no information to coordinated public information campaigns (including both traditional campaigns and those enacted via social media).
14. **Testing policy (H2)** relates to the varying levels of accessibility to no-cost COVID-19 testing, ranging from no testing policies to testing being open to the public (even to asymptomatic people).
15. **Contact tracing (H3)** relates to the varying levels of contact tracing after a case is confirmed, ranging from no contact tracing to contact tracing on all confirmed cases.
16. **Emergency investment in healthcare (H4)** includes the amount of money (USD) spent short-term on healthcare (e.g., temporary hospitals, masks, extra hospital supplies).
17. **Investment in vaccines (H5)** relates to the amount of money (USD) spent to help develop a COVID-19 vaccine.
18. **Facial coverings (H6)** relates to the varying levels of facial covering usage by individuals outside of their home, ranging from no policy to always required outside of homes with no exceptions.
19. **Vaccination policy (H7)** relates to the varying levels of vaccine availability and delivery for different groups of people, ranging from no availability to universal availability.
20. **Protection of the elderly (H8)** relates to the varying levels of care services and assistance (as defined locally) for older adults and their caregivers living in Long-Term Care Facilities, as well as community and home settings, to protect them against COVID-19 infection and its effects. Levels range from no measures in place to extensive restrictions for isolation and hygiene with no external visitors and “shelter-in-place” in effect, with minimal exceptions.

**Table 4 ijerph-20-07023-t004:** Days from first COVID-19 case until mitigation.

Mitigation	Bangladesh	Barbados	Belgium	Bolivia	Denmark	Estonia	Finland	Germany	Iceland	Lithuania	New Zealand	Norway	Serbia	Taiwan
School closing	22	16	53	15	29	32	63	44	31	8	36	29	24	23
Workplaces closing	25	25	52	21	27	43	57	69	31	11	36	27	24	126
Canceling public events	22	14	49	15	22	28	57	47	31	7	31	41	23	55
Restrictions on gatherings	25	14	57	15	29	41	57	57	30	7	31	28	19	126
Closing public transport	25	36	73	19	26	N/A	N/A	322	40	11	38	27	29	241
Stay-at-home requirements	25	25	57	20	19	45	61	56	N/A	11	36	267	23	130
Restrictions on internal movement	25	31	53	24	29	30	61	65	N/A	11	36	33	26	126
International travel controls	−32	−37	43	16	19	28	22	46	−16	8	−12	31	−30	N/A
Income support	50	29	45	34	25	17	61	63	36	33	32	37	39	102
Debt/contract relief	25	29	45	34	65	29	142	79	25	12	39	N/A	39	60
Fiscal measures	31	19	57	34	29	32	64	70	36	11	32	33	N/A	46
International support	28	108	N/A	N/A	49	59	92	112	N/A	−9	N/A	43	N/A	N/A
Public information campaigns	−33	−41	7	13	14	28	12	11	−22	−8	−23	−12	4	−8
Testing policy	7	−37	40	41	27	28	42	14	24	−36	14	14	6	10
Contact tracing	−30	14	36	97	14	189	44	9	24	33	14	49	14	11
Emergency investment in healthcare	24	240	59	62	49	63	176	70	N/A	N/A	32	2	39	48
Investment in vaccines	N/A	N/A	N/A	N/A	N/A	N/A	N/A	N/A	N/A	11	N/A	N/A	N/A	N/A
Facial coverings	97	39	94	65	147	85	211	79	168	155	180	184	131	14
Vaccine policy	295	−16	342	338	318	318	347	349	319	297	371	310	313	437
Protection of the elderly	−28	14	50	−56	33	28	61	46	21	7	40	295	23	115

N/A stands for no associated data.

**Table 5 ijerph-20-07023-t005:** Total duration (in days) of mitigation.

Mitigation	Bangladesh	Barbados	Belgium	Bolivia	Denmark	Estonia	Finland	Germany	Iceland	Lithuania	New Zealand	Norway	Serbia	Taiwan
School closing	564	561	566	568	548	559	562	583	564	529	190	568	564	411
Workplaces closing	510	552	567	373	569	466	568	558	468	483	175	564	533	139
Canceling public events	556	563	570	379	553	517	565	580	535	468	205	404	565	317
Restrictions on gatherings	553	467	562	344	537	555	507	570	536	514	205	520	528	139
Closing public transport	502	424	16	340	214	N/A	N/A	305	46	177	97	317	94	21
Stay-at-home requirements	553	394	395	406	421	109	375	409	N/A	352	118	324	530	69
Restrictions on internal movement	474	161	101	367	80	55	74	462	N/A	264	128	392	310	139
International travel controls	618	614	576	567	577	567	603	581	611	567	607	566	533	N/A
Income support	31	548	574	539	547	242	564	564	559	542	563	560	549	528
Debt/contract relief	463	548	574	355	272	422	483	91	570	563	556	N/A	535	448
Fiscal measures	3	2	11	5	5	10	4	3	3	10	2	13	N/A	3
International support	2	1	N/A	N/A	2	2	2	3	N/A	4	N/A	2	N/A	N/A
Public information campaigns	619	618	612	570	582	568	613	616	617	583	618	609	584	638
Testing policy	579	600	579	542	569	551	583	613	571	611	581	583	582	620
Contact tracing	544	543	583	480	582	398	581	618	571	542	581	548	574	619
Emergency investment in healthcare	2	1	2	7	2	2	1	3	N/A	N/A	1	4	3	2
Investment in vaccines	N/A	N/A	N/A	N/A	N/A	N/A	N/A	N/A	N/A	1	N/A	N/A	N/A	N/A
Facial coverings	489	538	525	518	449	511	414	548	417	420	415	407	457	616
Vaccine policy	282	227	277	245	278	278	278	278	276	278	224	287	275	193
Protection of the elderly	201	469	569	482	563	504	564	581	430	568	346	302	529	515

N/A stands for no associated data.

**Table 6 ijerph-20-07023-t006:** Wilcoxon rank-sum results regarding mitigation status: number of new cases per million.

Mitigation	Bangladesh	Barbados	Belgium	Bolivia	Denmark	Estonia	Finland	Germany	Iceland	Lithuania	New Zealand	Norway	Serbia	Taiwan
School closing	S	S	S	S	S	S	S	S	S	S	S	S	S	NS
Workplaces closing	S	S	S	S	S	S	S	S	S	S	S	S	S	S
Canceling public events	S	S	S	S	S	S	S	S	NS	S	S	S	S	S
Restrictions on gatherings	S	S	S	S	S	S	S	S	NS	S	S	S	S	S
Closing public transport	S	NS	S	S	S	N/A	N/A	S	NS	S	S	S	S	NS
Stay-at-home requirements	S	S	S	NS	S	NS	S	S	N/A	N/A	S	S	S	S
Restrictions on internal movement	S	S	S	NS	S	S	S	S	N/A	N/A	S	S	S	S
International travel controls	S	S	S	S	S	S	S	S	S	S	S	NS	S	N/A
Income support	S	S	S	S	S	S	S	S	S	S	S	S	S	S
Debt/contract relief	NS	S	S	S	NS	S	S	S	S	S	S	N/A	S	S
Fiscal measures	NS	NS	NS	S	NS	NS	NS	NS	NS	S	NS	S	N/A	NS
International support	NS	NS	N/A	N/A	NS	NS	NS	NS	N/A	N/A	N/A	NS	N/A	N/A
Public information campaigns	S	S	S	S	S	S	S	S	S	S	S	S	S	NS
Testing policy	S	S	S	S	S	S	S	S	S	S	S	S	S	S
Contact tracing	S	S	S	S	S	S	S	S	S	S	S	S	S	S
Emergency investment in healthcare	NS	NS	NS	S	NS	NS	NS	NS	N/A	N/A	NS	NS	NS	NS
Investment in vaccines	N/A	N/A	N/A	N/A	N/A	N/A	N/A	N/A	N/A	N/A	N/A	N/A	N/A	N/A
Facial coverings	S	S	NS	S	S	S	S	S	S	S	S	S	S	S
Vaccine policy	S	S	S	S	S	S	S	S	NS	S	S	S	S	S
Protection of the elderly	S	NS	S	S	S	NS	S	S	NS	S	NS	S	S	S

N/A stands for no associated data, S stands for significant difference between means, and NS stands for no significant difference between means.

**Table 7 ijerph-20-07023-t007:** Wilcoxon rank-sum results regarding mitigation status: viral reproduction rate.

Mitigation	Bangladesh	Barbados	Belgium	Bolivia	Denmark	Estonia	Finland	Germany	Iceland	Lithuania	New Zealand	Norway	Serbia	Taiwan
School closing	S	S	S	S	S	S	S	S	S	S	S	S	S	S
Workplaces closing	S	S	S	S	S	S	S	S	NS	S	S	S	S	NS
Canceling public events	S	S	S	S	S	S	S	S	S	NS	S	S	S	S
Restrictions on gatherings	S	S	S	S	S	S	NS	S	S	S	S	S	S	NS
Closing public transport	S	NS	NS	S	S	N/A	N/A	S	NS	S	S	S	S	S
Stay-at-home requirements	S	NS	S	S	S	S	S	NS	N/A	N/A	S	NS	S	S
Restrictions on internal movement	S	S	S	S	S	S	S	S	N/A	N/A	S	NS	S	NS
International travel controls	S	S	S	S	S	S	S	S	S	S	S	S	NS	N/A
Income support	S	S	S	S	S	S	S	S	S	S	S	S	S	S
Debt/contract relief	S	S	S	S	S	S	S	S	S	S	S	N/A	S	S
Fiscal measures	S	NS	NS	S	NS	NS	NS	NS	NS	NS	NS	S	N/A	NS
International support	NS	NS	N/A	N/A	NS	NS	NS	NS	N/A	N/A	N/A	NS	N/A	N/A
Public information campaigns	S	S	S	S	S	S	S	S	S	S	S	S	S	S
Testing policy	S	S	S	S	S	S	S	S	S	S	S	S	S	S
Contact tracing	S	S	S	NS	S	S	S	S	S	S	S	S	S	S
Emergency investment in healthcare	NS	NS	NS	NS	NS	NS	NS	NS	N/A	N/A	NS	S	NS	NS
Investment in vaccines	N/A	N/A	N/A	N/A	N/A	N/A	N/A	N/A	N/A	N/A	N/A	N/A	N/A	N/A
Facial coverings	S	S	S	S	S	S	S	S	S	S	S	S	S	S
Vaccine policy	NS	S	NS	NS	S	NS	NS	S	NS	NS	S	NS	NS	S
Protection of the elderly	S	NS	S	NS	S	S	S	S	S	S	NS	NS	S	S

N/A stands for no associated data, S stands for significant difference between means, and NS stands for no significant difference between means.

## Data Availability

The data presented in this study are openly available inhttps://github.com/OxCGRT/covid-policy-tracker (accessed on 19 October 2023).

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
