# Peer review of "Female Leadership during COVID-19: The Effectiveness of Diverse Approaches towards Mitigation Management during a Pandemic"

_ijerph, 2023, doi:10.3390/ijerph20217023_

Round 1
Reviewer 1 Report (Previous Reviewer 1)
Comments and Suggestions for Authors
With the revision done, the manuscript “Female Leadership during Covid-19: The effectiveness of diverse approaches towards mitigation management during a pandemic” is now much better. It should therefore be accepted for publication.
However, some improvements should still be made to the theoretical introduction and the conclusion.
The theoretical introduction now seems much better in terms of content, but subheadings should be inserted to make the various existing sections clearer. For example, it seems to me that at the end of the first paragraph a first subheading should be inserted to separate the initial brief theoretical introduction from the rest of the theoretical framework. This first subheading would cover the literature on feminist leadership theories. And then another subtitle should be inserted to cover the rest of the text, which is more practically orientated.
The various sections have been greatly improved by the revision, namely the Conclusion, which now includes the positioning of the female leaders analysed in relation to feminist leadership theories. However, the Conclusion now seems too long (four pages!), making it a bit tedious and monotonous to read. I think this problem could be solved with a more succinct conclusion, centred on the major findings of this study and its contributions to science.
Author Response
we have added a subheading after the first para that covers the literature on feminist leadership theories and then another subheading on summary of research findings on gender differences in COVID-19 management.
we have substantially reduced the conclusions and included a more succinct conclusion centered on our major findings and contributions to science.
Reviewer 2 Report (Previous Reviewer 2)
Comments and Suggestions for Authors
Dear Editors,
I endorse the reviewed article, as all the corrections and improvements have been well introduced. The paper provides a comprehensive analysis of how female leaders at the national level managed the COVID-19 response and recovery in their respective countries. The research is structured logically and includes a wide range of relevant variables, ensuring a comprehensive exploration of the topic. Additionally, the emphasis on qualities of leadership and their impact on trust and policy adherence is a noteworthy contribution to the field.
Author Response
Thank you for your time, dedication and expertise in reviewing our publication.
Reviewer 3 Report (Previous Reviewer 3)
Comments and Suggestions for Authors
Many thanks to the authors for the significant changes made to their text.
1. The graphics have been significantly improved, making them more readable.
2. The theoretical framework has been considerably expanded.
3. The selection of the country mandataries was explained.
4. The methodological explanation has been improved.
5. The discussion of the results was deepened.
6. References were updated considering more texts
In general, I consider that this version has what is necessary to be publishable.
Author Response
Thank you for your time, dedication and expertise in reviewing our publication.
Reviewer 4 Report (Previous Reviewer 4)
Comments and Suggestions for Authors
Thanks for improving the manuscript and responding satisfactorily to all of my earlier review comments. The manuscript's academic rigor has been enhanced. Congratulations.
Author Response
Thank you for your time, dedication and expertise in reviewing our publication.
This manuscript is a resubmission of an earlier submission. The following is a list of the peer review reports and author responses from that submission.
Round 1
Reviewer 1 Report
Comments and Suggestions for Authors
The manuscript “Female Leadership during Covid-19: The effectiveness of diverse approaches towards mitigation management during a pandemic” focuses on an important recent theme and is written in a very clear way. Therefore, I think it should be considered for publication.
I just share some comments and suggestions for improvement.
In section 3. Materials and Methods, I prefer to see the usual four subsections (Participants, Procedure, Instruments/Material, and Data Analysis) and, if the authors followed my suggestion, section 2. Female-led countries could be shifted from the Introduction and integrated in section 3. But as it stands the section is also quite clear.
It is not easy to understand the information of all the figures, because they are very small – I strongly suggest its enlargement.
From page 10, Table 4, the pagination of the text is completely changed.
I found it strange that in the Discussion, section 5.1. the authors referred so often to reference number 27 (I counted 20 times until the end of this section). I wonder if this is really so or if it is a mistake in the references.
Finally, I must say that, although I think the manuscript is written in an interesting and clear way, considering the title and the keywords, I was expecting to see a text more engaged in a gender perspective as the topic under analysis deserved (e.g., referring to what gender theories say about, for example, whether there are gender differences in leadership, or whether leadership is gendered or not, et cetera).
Author Response
Response to Reviewer 1
Comments and Suggestions for Authors
The manuscript “Female Leadership during Covid-19: The effectiveness of diverse approaches towards mitigation management during a pandemic” focuses on an important recent theme and is written in a very clear way. Therefore, I think it should be considered for publication.
I just share some comments and suggestions for improvement.
In section 3. Materials and Methods, I prefer to see the usual four subsections (Participants, Procedure, Instruments/Material, and Data Analysis) and, if the authors followed my suggestion, section 2. Female-led countries could be shifted from the Introduction and integrated in section 3. But as it stands the section is also quite clear.
Our analysis is at country level, so we are not reluctant to use participants. We don’t have instruments or materials we use strictly digital data and databases. So, we included your preferences of procedure, data and data analysis in section 3 and excluded participants and instruments/materials as subsections.
We kept section 2 as is and did not move.
It is not easy to understand the information of all the figures, because they are very small – I strongly suggest its enlargement.
We enlarged the figures.
From page 10, Table 4, the pagination of the text is completely changed.
I found it strange that in the Discussion, section 5.1. the authors referred so often to reference number 27 (I counted 20 times until the end of this section). I wonder if this is really so or if it is a mistake in the references.
We have used the World Factbook and referred to it for facts about the countries in our
manuscript. So it is not a mistake. We have included it in our reference list, see below.
Central Intelligence Agency. The World Factbook. Available online: https://www.cia.gov/theworld-factbook/ (Accessed 4 January 2023).
Finally, I must say that, although I think the manuscript is written in an interesting and clear way, considering the title and the keywords, I was expecting to see a text more engaged in a gender perspective as the topic under analysis deserved (e.g., referring to what gender theories say about, for example, whether there are gender differences in leadership, or whether leadership is gendered or not, et cetera).
We have added text more engaged in a gender perspective and referred to what female leadership theories say about gender differences in leadership, or whether leadership is gendered or not. Our manuscript is focused on comparison of female leaders during COVID and their mitigation strategies. With this paper, we determined which COVID-19 mitigations were effective in Female Presidents’ countries led by female Presidents. The paper aimed at showing whether leaders' gender and their traits affected the way the countries dealt with Covid-19. Our manuscript complements the research published on gender differences in leadership and further validates how female leaders managed and controlled COVID in their respective countries.
Reviewer 2 Report
Comments and Suggestions for Authors
The paper deals with and important issue of whether determine which COVID-19 mitigations were effective inFemale Presidents’ countries led by female Presedents of Prime Ministers. The paper aimed at showing whether leaders' gender affected the way the countries dealt with Covid-19 which is a very interesting topic. The conclusions of teh article should be more clearly stated. They are a bit vague. Moreover, the question the writers ask in the conclusion - "Why should the qualities of community-based, empathetic, and personable leadership, which prioritize human lives and prove to be sustainable in the long run, be limited to the "feminine domain"? is quite contraictory to their initial research question wityh regard to gender's effect on female leaders' success in dealing with Covid. If gender is irrelevant or if, as the writer state, gender is"deeply ingrained patriarchal beliefs rooted in biological determinism" - why then they base their research on gender differences?
Author Response
Response to Reviewer 2
Comments and Suggestions for Authors
The paper deals with and important issue of whether determine which COVID-19 mitigations were effective in Female Presidents’ countries led by female Presidents of Prime Ministers. The paper aimed at showing whether leaders' gender affected the way the countries dealt with Covid-19 which is a very interesting topic. The conclusions of the article should be more clearly stated. They are a bit vague. Moreover, the question the writers ask in the conclusion - "Why should the qualities of community-based, empathetic, and personable leadership, which prioritize human lives and prove to be sustainable in the long run, be limited to the "feminine domain"? is quite contradictory to their initial research question with regard to gender's effect on female leaders' success in dealing with Covid. If gender is irrelevant or if, as the writer state, gender is "deeply ingrained patriarchal beliefs rooted in biological determinism" - why then they base their research on gender differences?
We worked on the conclusions of the article and made it clearer. We have added text more engaged in a gender perspective and referred to what female leadership theories say about gender differences in leadership. We removed the contradictory questions and statements in the conclusion to our initial research question. We like to emphasize that our manuscript is focused on comparison of female leaders during COVID and their mitigation strategies. With this paper, we determined which COVID-19 mitigations were effective in Female Presidents’ countries. Our manuscript complements the research published on gender differences in leadership during COVID and further validates how female leaders effectively managed and controlled COVID in their respective countries. We concluded that understanding what they did to make it so effective could be beneficial for other countries who did not do that well. Our findings contributed to the understanding of female leaders' success in dealing with Covid at its early stages and show that Women can be effective leaders in crisis and management. We included in future implications that we will address differences in leadership between genders and their effectiveness of mitigation management.
Reviewer 3 Report
Comments and Suggestions for Authors
Overall the article appears to be interesting, as it purports to show a relationship between having a female president and the response to the pandemic. However, I am concerned that the results are not conclusive, and thus leave the objective half-finished:
1. the entire article revolves around the fact that the sample is constructed from countries with female presidents. However, when the analysis is carried out, it is clear that there are no common parameters, but rather that there are countries with good performance and some with poor performance. This calls into question the idea of gender as a conclusive element, but rather, that the situation of each country may respond to elements not necessarily associated with gender. This considerably minimizes the value of the article.
I think the idea is good, but I would need to expand the studies to be able to demonstrate a statistically significant correlation between the variables associated with attention to COVID and the fact of having or not having a female president. For this, I would also expect a comparison with other nations with male presidents, to know if this is truly an impacting element.
2. I am concerned that most of the references used are from 2020 results, which was a year in which we were still in a pandemic. I consider that the true results of the situation should be taken after 2022, since in 2020 there was neither complete information nor clear information. In this sense, I believe that it only shows us half information.
3. The tables and figures cannot be appreciated correctly. The tables are too big, the figures too small.
4. The conclusions lack a section on limitations and possible derivative studies.
Author Response
Response to Review 3
Comments and Suggestions for Authors
Overall the article appears to be interesting, as it purports to show a relationship between having a female president and the response to the pandemic. However, I am concerned that the results are not conclusive, and thus leave the objective half-finished:
- the entire article revolves around the fact that the sample is constructed from countries with female presidents. However, when the analysis is carried out, it is clear that there are no common parameters, but rather that there are countries with good performance and some with poor performance. This calls into question the idea of gender as a conclusive element, but rather, that the situation of each country may respond to elements not necessarily associated with gender. This considerably minimizes the value of the article.
In response to your valuable feedback, we made the necessary adjustments in the article to explicitly emphasize that our findings do not assert gender as the sole or conclusive element influencing outcomes though we found common parameters associated with female leadership and we added the frequency of each mitigation that was found important throughout the models. We recognized that a comprehensive analysis should consider multiple factors, and we appreciate your insights on this matter.
I think the idea is good, but I would need to expand the studies to be able to demonstrate a statistically significant correlation between the variables associated with attention to COVID and the fact of having or not having a female president. For this, I would also expect a comparison with other nations with male presidents, to know if this is truly an impacting element.
The challenges faced and decisions made in 2020 were crucial in shaping the subsequent phases of the pandemic response. However, we value your input and will work on addressing your concerns by considering additional data from the post-2020 period in our future research and work on a future article comparing with other nations with male presidents and different time periods in the beginning versus throughout. We hope that this response clarifies our approach and intent.
- I am concerned that most of the references used are from 2020 results, which was a year in which we were still in a pandemic. I consider that the true results of the situation should be taken after 2022, since in 2020 there was neither complete information nor clear information. In this sense, I believe that it only shows us half information.
We are committed to addressing your concerns regarding the validity of our study period and the choice of references used from 2020 results. First and foremost, we understand your concern about the predominance of references from the year 2020, a period marked by the global pandemic. It is indeed a valid point to consider that the pandemic situation was evolving rapidly during that year, and complete and clear information was sometimes challenging to obtain. However, we would like to provide some clarification regarding our choice of study period.
Our primary objective was to analyze the strategies implemented by female leaders during the peak of the pandemic, which includes the year 2020. We believe that assessing the immediate responses of female leaders during the initial stages of the pandemic offers valuable insights into crisis management and leadership. This way we found out which female leaders did timely and efficient management and control of the pandemic and reduced the number of cases and deaths and the spread of the disease during a time immediate action with limited knowledge and resources.
- The tables and figures cannot be appreciated correctly. The tables are too big, the figures too small.
We enlarged the figures and minimized the tables.
- The conclusions lack a section on limitations and possible derivative studies.
We have included a section on limitations in our conclusion and we added derivative studies.
Reviewer 4 Report
Comments and Suggestions for Authors
1. The abstract must include the methods for the study which has not been well articulated in the section.
2. While the background to the study is theoretically sound, there is the need to review feminist theories of leadership to enhance the theoretical bedrock for the study.
3. Under the section 'Female-led countries', instead just stating the names of the female leaders who served as chief executives for the governance of their respective countries, it would be good.
4. The first section of the 3.2 Methods which is the limitations must be moved from the section to the concluding section. The section must show the study approach and design first before the analytical procedures used for the treatment of the data.
5. A more scholarly justification for the use of the random forest modeling/analysis must be given from the literature. Why was this analytical procedure more appropriate for this study?
6. The results and discussions have been scholarly presented. However, there is the need to introduce a section on implrcations of the study for policy guidelines in terms of effective mitigation strategies that could be used in case of any future health emergency.
7. What areas needed to be heightened by the Female-led governments (if any). This could help those countries to strengthen their mitigation support structures as well.
8. The concluding section needs to be re-written. The current version could be used as a starting point for policy implications (the needed section I earlier suggested). It should be noted that a good concluding section is supposed to:
1. Summarize the purpose of the study and its key or principal findings
2. Tentative inferences and conclusions drawn solely from the key findings
3. Limitations of the study
4. Recommendations for theory and policy
5. Recommendations for future research
No literature discussions should be done in the concluding section.
Author Response
Response to Review 4
Comments and Suggestions for Authors
- The abstract must include the methods for the study which has not been well articulated in the section.
We included the methods for the study in the abstract. We provided a more scholarly justification for the use of the random forest modeling/analysis from the literature. We also provided explanation why this analytical procedure was more appropriate for this study. We demonstrated the utility of the random forest approach on the predictive significance of these variables providing more interpretability.
- While the background to the study is theoretically sound, there is the need to review feminist theories of leadership to enhance the theoretical bedrock for the study.
We added a review of feminist theories of leadership and enhanced the theoretical bedrock for the study.
- Under the section 'Female-led countries', instead just stating the names of the female leaders who served as chief executives for the governance of their respective countries, it would be good.
Under the section 'Female-led countries’, we stated the names of the female leaders who served as chief executives for the governance of their respective countries.
- The first section of the 3.2 Methods which is the limitations must be moved from the section to the concluding section. The section must show the study approach and design first before the analytical procedures used for the treatment of the data.
We moved the first section of the 3.2 Methods to the concluding section under the limitations. We added “Procedure” section and first introduced the study approach and design in the Procedure section before data and data analysis sections where the analytical procedures used for the treatment of the data were written in detail.
- A more scholarly justification for the use of the random forest modeling/analysis must be given from the literature. Why was this analytical procedure more appropriate for this study?
We provided a more scholarly justification for the use of the random forest modeling/analysis from the literature. We also provided explanation why this analytical procedure was more appropriate for this study. We demonstrated the utility of the random forest approach on the predictive significance of these variables providing more interpretability.
- The results and discussions have been scholarly presented. However, there is the need to introduce a section on implications of the study for policy guidelines in terms of effective mitigation strategies that could be used in case of any future health emergency.
We introduce a section titled ‘future implications of the study’ on implications of the study for policy guidelines in terms of effective mitigation strategies that could be used in case of any future health emergency.
- What areas needed to be heightened by the Female-led governments (if any). This could help those countries to strengthen their mitigation support structures as well.
We included the areas needed to be heightened by the Female-led governments so that this could help those countries to strengthen their mitigation support structures as well. we included areas that could benefit incorporating policies, mitigation strategies, better communication as those led by female led governments in the future policy implications subsection of the conclusions.
- The concluding section needs to be re-written. The current version could be used as a starting point for policy implications (the needed section I earlier suggested). It should be noted that a good concluding section is supposed to:
We re-wrote the conclusion. We added limitations and future policy implications sections. In conclusions, we summarized the purpose of the study and its key or principal findings; referred to feminist leadership theories; tentative inferences and conclusions drawn solely from the key findings; Limitations of the study; Recommendations for theory and policy; Recommendations for future research.
- Summarize the purpose of the study and its key or principal findings
done
- Tentative inferences and conclusions drawn solely from the key findings
done
- Limitations of the study
added
- Recommendations for theory and policy
done
- Recommendations for future research
done
No literature discussions should be done in the concluding section.
done
Reviewer 5 Report
Comments and Suggestions for Authors
The article can be interesting, since it tries to show the management of the covid by women.
Although an inferential comparison of the data is necessary to establish this assertion, statistical tests are necessary to support the results.
Therefore, the authors must make a new statistic, and reassess the results.
Author Response
Response to Review 5
Comments and Suggestions for Authors
The article can be interesting, since it tries to show the management of the covid by women.
Although an inferential comparison of the data is necessary to establish this assertion, statistical tests are necessary to support the results.
Therefore, the authors must make a new statistic, and reassess the results.
We applied the Wilcoxon rank-sum statistical test to see the differences with and without mitigation in effect for the variables that were found significant by the random forest model. We added justification for the use of random forest predictive model. We added the new statistic as subsection under methods and materials. We also added the results of the statistic to the results section. We reassessed our results and included in our conclusions.